# The Effect of Antibiotic Treatment and Gene Expression of Mex B Efflux Transporters on the Resistance in *Pseudomonas* Aeruginosa Biofilms

Evan Kello [1,†], Rochelle Greenberg [1,†], Weiqi Li [2], Shaya Polansky [3], Roberto Maldonado [2], Yakov Peter [3] and Paramita Basu [1,2,*]

1   New York College of Podiatric Medicine, 53 E 124 Street, New York, NY 10035, USA; ekello2026@nycpm.edu (E.K.); rgreenberg2026@nycpm.edu (R.G.)
2   Touro College of Pharmacy, 230 W 125 Street, New York, NY 10027, USA; wli11@student.touro.edu (W.L.); rmaldona2@student.touro.edu (R.M.)
3   Lander College for Men, Touro University, 75-31 150th Street, Flushing, NY 11367, USA; shayapolansky@gmail.com (S.P.); yakov.peter@touro.edu (Y.P.)
*   Correspondence: paramita.basu@touro.edu
†   These authors contributed equally to this work.

**Abstract:** *Pseudomonas aeruginosa*, a Gram-negative, rod-shaped bacterium, holds a prominent position as an antibiotic-resistant priority pathogen, according to the World Health Organization. Particularly prevalent in healthcare settings, this bacterium acts as an opportunistic pathogen, causing nosocomial infections. The significant antibiotic resistance observed in *P. aeruginosa* is multifactorial, encompassing intrinsic, acquired, and adaptive resistance mechanisms. The present study aims to explore specific RND-type efflux pump genes implicated in the acquisition of antibiotic resistances during the transition of *P. aeruginosa* PAO1 from its planktonic state to the more formidable and resistant biofilm form. This investigation is centered on MexB, a prominent RND-type efflux pump in P. *aeruginosa*. Our research is focused on MexB, a highly significant component characterized by its broad substrate specificity, primary function as the primary efflux pump, substantial expression levels, and notable clinical implications. Considering MexB's critical role in expelling various clinically relevant antimicrobial agents and its significant contribution to multidrug resistance, our study aims to evaluate the comparative efficacy of three distinct antibiotic categories, namely, Ofloxacin (OFX), Tobramycin (TOB), and Ceftazidime (CAZ), in regulating the expression levels of identified multidrug efflux pump genes associated with the biofilm's ability to remove antibiotics from bacterial cells. Expression analysis of efflux transporter genes in *P. aeruginosa* was performed by isolating total RNA from both planktonic and biofilm samples, both untreated and treated with Tobramycin (TOB), Ofloxacin (OFX), and Ceftazidime (CAZ). Real-time quantitative reverse transcriptase PCR was employed to investigate changes in the expression levels of MexA, MexB, MexX, MexY, OprM, and RPSL genes in the collected samples. In the absence of antibiotic treatment, the MexB efflux pump gene exhibited higher expression compared to other efflux pump genes in the biofilm's state, supporting its involvement in multidrug resistance when active. To further explore the role of the MexB gene in antibiotic resistance, *P. aeruginosa* was cultured in both planktonic and biofilm forms while simultaneously treating them with TOB, OFX, and CAZ. Among the three antibiotics employed, OFX demonstrated superior efficacy in inhibiting the growth of biofilms by downregulating the expression of the Mex B efflux pump gene in *P. aeruginosa*, thereby enhancing its susceptibility to OFX. TOB yielded comparable outcomes to OFX, albeit with a slightly lesser extent of Mex B expression reduction. Conversely, CAZ exhibited ineffectiveness in reducing MexB gene expression in both biofilm and planktonic forms of the organism, rendering it incapable of eradicating the pathogen.

**Keywords:** antibiotic resistance; biofilm; pseudomonas aeruginosa; ofloxacin; tobramycin; ceftazidime; antibiotic sensitivity; efflux transporters

## 1. Introduction

*Pseudomonas aeruginosa* is a Gram-negative, rod-shaped bacterium commonly found in healthcare facilities, where it poses a significant threat as an opportunistic pathogen causing nosocomial infections associated with high mortality rates [1]. *P. aeruginosa* can cause both acute and chronic infections, each following distinct disease courses. Acute infections, which are generally more manageable, typically involve *P. aeruginosa* in its planktonic form and can be effectively treated with antibiotics. On the other hand, chronic infections are characterized by the formation of antibiotic-resistant biofilms by *P. aeruginosa*. Biofilms offer an optimal milieu for the dissemination of *P. aeruginosa* infections. Once established, these biofilms have the capacity to liberate planktonic cells into the surrounding milieu, thereby facilitating colonization at new anatomical sites within the host or potentially contaminating medical apparatus and surfaces within healthcare facilities [2]. Consequently, this perpetuates the persistence of the infection and augments the susceptibility of vulnerable individuals, such as immunocompromised patients, to the risk of transmission. *P. aeruginosa* biofilms are notorious for their ability to develop multi-drug antibiotic resistance (MDR), posing a considerable challenge in eradicating infections, leading to prolonged illnesses and increased healthcare costs [3].

*P. aeruginosa* cells within biofilms exhibit increased virulence compared to their planktonic counterparts. This enhanced virulence is essential to their ability to coordinate behavior, engage in quorum-sensing [2–4] communication, and express virulence factors that facilitate tissue damage and evade the host's immune system. In the context of biofilms, efflux pumps indirectly contribute to the virulence of *P. aeruginosa* by influencing the overall fitness and persistence of the biofilm community. Efflux pumps are membrane-associated proteins that expel substances from bacterial cells, including antibiotics, waste products, and toxins. In specific, the RND (Resistance–Nodulation–Division)-type multidrug resistance efflux pumps play a crucial role in this process by enabling broad substrate specificity, contributing to both intrinsic and acquired resistance mechanisms, promoting the formation and persistence of biofilms, and impacting the effectiveness of treatment approaches [5]. The presence of these efflux pumps poses a significant challenge in managing multidrug-resistant *P. aeruginosa* infections, emphasizing the need for a comprehensive understanding of their underlying mechanisms. Moreover, exploring the development of efflux pump inhibitors holds promise as a potential adjunctive therapy to address the complexities associated with *P. aeruginosa* infections, particularly those involving multidrug-resistant strains [5–7].

RND efflux pump systems associated with *P. aeruginosa* are mainly the following: MexAB-OprM, MexCD-OprJ, MexEF-OprN, and MexXY-OprM [8]. The RND-type efflux pumps account for the major cause of intrinsic resistance to most antimicrobial agents in *P. aeruginosa* [9]. MexXY-OprM and MexAB-OprM are known to be the largest and only intrinsic multidrug-resistant efflux pumps within the resistance nodulation division (RND) family in *P. aeruginosa* [10]. The pumps extrude antimicrobials across the outer membrane, conferring resistance to beta-lactams targeting cell wall synthesis. MexAB-OprM consists of three different peptides: a MexB translocase solute/proton RND antiporter, an outer membrane porin-OprM, and a membrane fusion protein MexA, which docks MexB to OprM. Studies suggest that the MexAB-OprM system is constitutively expressed and contributes to the efflux of a broad range of antibiotics. In contrast, the MexXY system is inducible and primarily involved in the efflux of only aminoglycosides [10]. Given the broad substrate specificity of the MexAB-OprM efflux pump system, MexB hyperexpression is pivotal to the intrinsic resistance of *P. aeruginosa* to several antimicrobials, including fluoroquinolones, tetracyclines, and β-lactams, such as carbapenem [11,12]. Biofilms demonstrate significantly elevated levels of resistance, and it remains uncertain whether the mechanisms that confer lower resistance in planktonic cells also play a significant role in biofilms [5]. Studies identified an upregulation of the MexAB-OprM efflux pump in the clinical biofilm isolates when compared to the control planktonic strain, as well as decreased expression of OprD, indicating a potential link between these expression levels

and antibiotic resistance [12]. This enhanced expression implies that *P. aeruginosa* strains employ this mechanism to actively expel antibiotics, thereby strengthening their resistance to antimicrobial agents. The observed MexB upregulation aligns with the commonly observed increased resistance correlated to biofilms, without the inclusion of antibiotic therapy [12,13]. Other studies suggest that MexB critically contributes to the simultaneous overexpression of MexAB-OprM and MexXY pumps in PAO1 clinical strains [14]. Genetic alterations in regulatory genes and promoters affect MexB expression and other pump components, increasing pump activity. This heightened dual expression, mediated by MexB, actively promotes the efflux of antibiotics, reducing their effectiveness and promoting the development of antibiotic resistance in *Pseudomonas aeruginosa* PAO1 [15]. While the association between MexB hyperexpression and untreated *P. aeruginosa* strains is supported, further investigation is required to unravel its specific role concerning different antibiotics and growth stage-specific interactions, such as planktonic versus biofilm expression.

This study aims to elucidate the role of the MexB gene in influencing the susceptibility of *P. aeruginosa* to antibiotic treatment. To accomplish this, the study employs different classes of antibiotics to assess the effect on bacterial susceptibility and its greater contribution to MDR. Reverse transcription polymerase chain reaction (RT-PCR) analysis is used to investigate any changes in activity or mutations in the MexB gene. The comparative analysis of the Minimum Inhibitory Concentration (MIC) and Minimum Bacterial Concentration (MBC) values of *P. aeruginosa*, both in the planktonic stage and biofilms with functional MexB, in the presence of various antibiotics, offers valuable insights into the role of MexB in antibiotic resistance. By exploring the significance of the MexAB-OprM efflux pump system, the principal broad-spectrum intrinsic pump, a better understanding of how MexB influences antibiotic resistance in biofilms can be achieved. This understanding can contribute to the development of more effective treatment strategies targeting MexB-mediated resistance in biofilms.

## 2. Materials and Methods

### 2.1. Literature Search

A literature review took place between August 2020 and December 2020, gathering studies related to current treatment approaches for *P. aeruginosa* and different efflux transporters. Promising antibiotics identified for treating *P. aeruginosa* included Tobramycin (TOB), Ofloxacin (OFX), and Ceftazidime (CAZ). The Basic Local Alignment Search Tool (BLAST) was utilized to identify regions of similarity in the genetic sequences of MexA, MexB, MexX, and MexY genes from the PAO1 strain. These sequences were compared to well-studied genes from various strains, which were extensively documented in the literature regarding their functionality under different stress conditions. BLAST is an indispensable tool for biologists, as it efficiently and sensitively compares nucleotide and protein sequences with both individual sequences and large databases, enabling the design of primers.

### 2.2. Planktonic Cell Growth

Modifications to the planktonic cell growth measurement assay based on the method described by Qu et al. in 2016 were performed. Initially, *P. aeruginosa* PAO1 cells that were grown overnight were diluted 1:100 in LB broth. These diluted samples were then incubated at 37 °C with continuous agitation at 160 rpm. To monitor the turbidity of the planktonic culture, we used an Eppendorf spectrophotometer to measure the absorbance at 550 nm at hourly intervals.

### 2.3. Determination of Minimal Bacterial Concentration (MBC) and Minimal Inhibition Concentration (MIC)

The minimal bacteriostatic concentration of an antimicrobial agent was determined using a 96-well plate. Each well contained 12 columns with 3 rows of antibiotics and the *P. aeruginosa* bacteria. To achieve minimal inhibition, 25 µL of antibiotic and 100 µL of diluted

bacteria were simultaneously added to each well. The antibiotics (TOB, OFX, and CAZ) were diluted to concentrations of 256 µg/mL, 128 µg/mL, 64 µg/mL, 32 µg/mL, 16 µg/mL, 8 µg/mL, 4 µg/mL, 2 µg/mL, 1 µg/mL, 0.5 µg/mL, 0.25 µg/mL, and a control. The final volume in each well was 125 µL. The samples were then incubated for 24 h at 37 °C.

For the minimum bactericidal concentration (MBC), 100 µL of PAO1 was added to each well and incubated for 24 h at 37 °C. Then, 25 µL of antibiotic was added to each well, totaling 125 µL, followed by a subsequent 24 h incubation at 37 °C. After incubation was complete for both plates, the plates were washed three times with deionized water and allowed to dry for 4–6 h. Subsequently, to test the sensitivity to specific antibiotic concentrations, 135 µL of a 0.1% solution of crystal violet was added to each well and incubated at room temperature for 10–15 min. Another three sets of rinses were performed, and the plate was left to dry overnight. To quantify the biofilm, 125 µL of a 30% acetic acid solution in water was added to each well to dissolve the crystal violet, followed by a 10–15 min incubation. The 96-well plate was then placed in a spectrophotometer to measure the absorbance (OD at 550 nm).

### 2.4. Viability Assay Using Spectrofluorometric Analysis

To investigate the antibacterial efficacy of the three different antibiotics (TOB, OFX, and CAZ) on the PAO1 stand of *Pseudomonas Aeruginosa*, we also examined the reduction in cell viability by employing Propidium Iodide (PI) fluorescence staining. PI is a fluorescent dye that cannot penetrate intact bacterial cell membranes. However, when the membrane integrity is compromised, PI can enter the cells and bind to their nucleic acids.

The PAO1 cells were cultivated and exposed to various concentrations of antibiotics as previously described. Subsequently, the cells were treated with 10 µM of Propidium iodide (PI). After incubating the cells with the stain for 10 min, the average fluorescence intensities resulting from the binding of PI to the cells were measured using a Molecular Devices SpectraMax Gemini® microplate spectrofluorometer, following the instructions provided in the BacLight™ LiveDead Kit (Molecular Probes, Eugene, OR, USA).

### 2.5. Gene Expression Analysis

After determining the minimal inhibitory concentration (MIC) and minimal bactericidal concentration (MBC), concentrations of 8 µg/mL and 32 µg/mL of TOB, OFX, and CAZ were selected for MIC and MBC evaluations, respectively. These concentrations were chosen based on optimal observation of inhibition (at 8 µg/mL) and eradication (at 32 µg/mL).

To analyze the expression of the efflux transporter genes, total RNA was extracted from the cells and treated with the selected concentrations using the RNeasy Mini kit (Qiagen, Hilden, Germany), following the kit instructions. Additionally, RNAprotect® Bacteria Reagent was added at the appropriate step, as indicated in the RNeasy kit instructions, to ensure RNA stability. The purity and concentration of the extracted RNA were determined by measuring the absorbance at 260 nm and 280 nm (260/280 nm ratio). For cDNA synthesis, 1 µL of RNA was used with the TransScript All-in-One First-Strand cDNA Synthesis SuperMix (Transgene, Beijing, China).

The expression of the efflux transporter genes was assessed by quantitative real-time PCR (qRT-PCR) amplification and quantification using the synthesized cDNA. The qRT-PCR was conducted using the TransStart™ Green qPCR SuperMix UDG kit (Transgene, Beijing, China). The qRT-PCR conditions consisted of an initial denaturation step at 94 °C for 10 min, followed by 40 cycles of amplification with 5 s at 94 °C and 30 s at 60 °C. The obtained data were normalized to the endogenous reference gene RPSL of the *P. aeruginosa* PAO1 strain. Changes in gene expression in each sample relative to the control were analyzed using the threshold cycle method ($2\hat{}(-\Delta\Delta CT)$). The qRT-PCR experiments were performed in triplicate using the ProtoScript® First Strand cDNA Synthesis Kit, and the entire experiment was repeated twice using RNA samples extracted from independent cultures.

## 3. Results and Discussion

### 3.1. Selection of Primers for Gene Expression Studies

The BLAST tool available on the National Library of Medicine website was utilized to align and compare the DNA sequences of the genes MexA, MexB, MexX and MexY. In the context of the biofilm state and in the absence of antibiotic treatment, the expression of the MexB efflux pump gene was found to be significantly upregulated relative to other efflux pump genes This distinctive upregulation, coupled with the fact that MexB is the sole intrinsic efflux pump possessing broad specificity [6], drew our focus to examining solely MexB as a significant player in efflux-mediated resistance mechanisms within the *P. aeruginosa*. MexB was selected for further studies on the effect of antibiotics on planktonic cells and biofilms formed by the *P. aeruginosa* PAO1 strain based on literature search to design the primers needed to facilitate the amplification and quantification of gene expression levels by a quantitative real-time polymerase chain reaction (qRTPCR).

The BLAST alignment tool was used to locate similar nucleotides between the biological sequences of the four efflux transporter genes MexA, MexB, MexX and MexY, from *P. aeruginosa* PAO1, with simultaneous comparisons to other strains and recorded genes.

We utilized qRT-PCR-based amplification and quantification of cDNA obtained from the transcribed RNA isolated from the tested samples. Specific primers targeting the four genes associated with efflux transporter pumps were sequenced. While each of these genes plays a vital role in actively expelling antibiotics from bacterial cells, MexB was selected for further analysis based on its broad substrate specificity [10,16], high expression levels [5,17–22], genetic regulation [5,6,16], and clinical relevance [5,22]. Our analysis focused on determining the relative expression levels of MexB and investigating its potential involvement in antibiotic extrusion and resistance in *P. aeruginosa* PAO1 [10,13,14].

The four potent RND-type multidrug resistance efflux pumps—MexA, MexB, MexX, and MexY—remove intracellular harmful chemicals in *P. aeruginosa*, including antibiotics. By diminishing the intracellular antibiotic concentration, leading to decreased susceptibility and increased resistance, the efflux pump systems play a direct role in the development of multidrug-resistant *P. aeruginosa.* (Table 1). The ribosomal subunit (RPSL) responsible for rRNA and tRNA binding is constitutively expressed and utilized as the internal control for monitoring changes in the efflux transporters' expression levels in response to biofilm formation and the addition of varied antibiotics [16–18].

### 3.2. Effectiveness of Different Antibiotics for Inhibition and Eradication of Pseudomonas Aeruginosa Biofilm

Figure 1A,B compares the effectiveness of three antibiotics, ceftazidime (CAZ), ofloxacin (OFX), and tobramycin (TOB), in inhibiting biofilm formation and eradicating mature biofilms in the *P. aeruginosa* PAO1. The Minimum Inhibitory Concentration (MIC) and Minimum Bactericidal Concentration (MBC) values of these antibiotics assess the antibiotic capacity to inhibit and eradicate biofilms. Figure 1B, the eradication data, evaluates the effectiveness of these antibiotics on mature biofilms, which mimic clinical biofilm infections. Comparatively, the inhibition data in Figure 1A investigate the effects of the mentioned antibiotics on biofilm formation originating from free-living or planktonic cells.

The impact of antibiotics on biofilm inhibition (n = 6), involving the application of the antibiotic during inoculation (a), as well as biofilm eradication (n = 6), where the antibiotic is applied after the formation of mature biofilms (b), was evaluated at different concentrations. The x-axis illustrates the different concentrations of the three antibiotics, whereas the y-axis represents the degree of inhibition or eradication of *P. aeruginosa* biofilms resulting from treatment with the antibiotics. The measurement of residual biofilms was conducted using spectrophotometric absorbance at 550 nm, providing a quantitative assessment of biofilm levels.

**Table 1.** Summary of the Functions of Efflux Transporters MexA, MexB, MexX, and MexY in Bacteria.

| Protein | Function | Role in Antibiotic Resistance | Primer Sequence |
|---|---|---|---|
| Multidrug Resistant Efflux Pump MexA | Resistance-Nodulation-Cell Division (RND) multidrug efflux Periplasmic membrane fusion protein precursor | Transports structurally varied molecules, including antibiotics, out of the bacterial cell | F: 5′-acctacgaggccgactaccaga-3′ R: 5′-gttggtcaccagggcgcctc-3′ |
| Multidrug Resistant Efflux Pump MexB | Resistance-Nodulation-Cell Division (RND) Inner membrane multidrug efflux Transporter protein | Transports structurally varied molecules, including antibiotics, out of the bacterial cell | F: 5′-gtgttcggctcgcagtactc-3′ R: 5′-aaccgtcgggattgaccttg-3′ |
| Outer Membrane Protein OprM | Major intrinsic multiple antibiotic resistance efflux outer membrane protein OprM precursor | Channel-forming outer membrane protein | F: 5′-ccatgagccgccaactgtc-3′ R: 5′-cctggaacgccgtctggat-3′ |
| Multidrug Resistant Efflux Pump MexX | Resistance-Nodulation-Cell Division (RND) multidrug efflux membrane fusion protein MexX precursor | Transports structurally varied molecules, including antibiotics, out of the bacterial cell | F: 5′-tgtacgcgtattcggaacaaggcgtctgc-3′ R: 5′-ttctgctagcgatgtgcatgggtgtccctc-3′ |
| Multidrug Resistant Efflux Pump MexY | Resistance-Nodulation-Cell Division (RND) multidrug efflux transporter MexY | Transports structurally varied molecules, including antibiotics, out of the bacterial cell | F: 5′-tgtactagttgatgcccctagcgaaactctc-3′ R: 5′-tttaagcttgacctacaggacgctgctg-3′ |
| Ribosomal Subunit RPSL | Ribosomal subunit binding rRNA and tRNA, expressed constitutively | Structural constituent of ribosome which serves as Internal control | F: 5′-gctgcaaaactgcccgcaacg-3′ R: 5′-acccgaggtggtccagcgaacc-3′ |

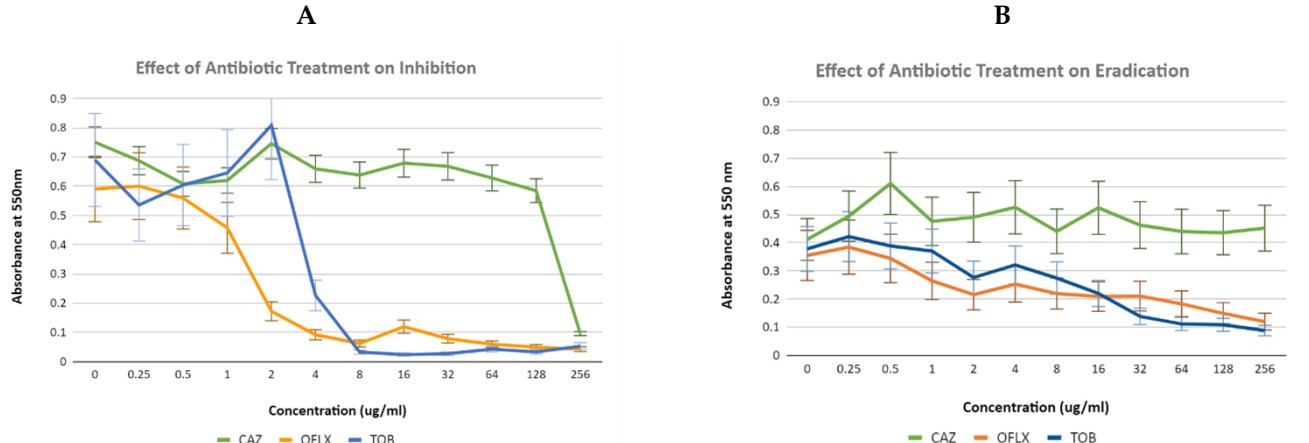

**Figure 1.** Effect of increasing concentration of the ceftazidime (CAZ), ofloxacin (OFLX), and tobramycin (TOB) on (**A**) inhibition and (**B**) eradication *of Pseudomonas aeruginosa* biofilm.

In order to compare the effectiveness of the three antibiotics in inhibiting *P. aeruginosa* biofilms, the impact of different concentrations of each antibiotic on biofilm formation capacity was evaluated using the microbroth format. This assessment aimed to determine the

Minimum Inhibitory Concentration (MIC) values. The results indicated a dose-dependent sensitivity of PAO1 biofilms to both ofloxacin and tobramycin at concentrations of 8 μg/mL while exhibiting high resistance to ceftazidime during the inhibition phase. In contrast, the efficacy of the three antibiotics in eradicating *P. aeruginosa* biofilms was assessed by evaluating the remaining biofilm following incubation with different concentrations of each antibiotic to determine the Minimum Bactericidal Concentration (MBC) values.

The results demonstrated notable resistance to all antibiotics, except for ofloxacin, when targeting mature biofilms. During the eradication phase, biofilms exhibited sensitivity to ofloxacin, but at significantly higher concentrations (32 μg/mL) compared to the inhibition phase. Additionally, biofilms showed sensitivity to tobramycin only at the highest concentration tested (256 μg/mL). Ceftazidime displayed the highest level of resistance, demonstrating insensitivity even at the highest concentration, and exhibited considerable resistance to tobramycin at lower concentrations during the eradication phase. For the inhibition phase (Figure 1A), antibiotics were administered at the time of inoculation in a 96-well microtiter plate. In the eradication phase (Figure 1B), antibiotics were applied 24 h after the 37 °C inoculation. At 550 nm, the optical density (OD) of each sample was measured following incubation with increasing antibiotic dosing. Overall, ofloxacin exhibited the greatest inhibitory and eradication effects, whereas ceftazidime demonstrated the least effectiveness (Figure 1A,B) in both conditions. These observations supported studies that suggested *P. aeruginosa* biofilms imparted resistance to beta-lactams (Ceftazidime), aminoglycosides (Tobramycin), and fluoroquinones (Ofloxacin) through the upregulation of efflux pumps [16,18]. In particular, the MexAB-OprM efflux pump system was reported to contribute to conferring inherent resistance to β-lactams and quinolones [17,18]. Biofilms will not be eradicated with low-dose tobramycin [19–21], but ofloxacin is supported as the choice of drug for *P. aeruginosa* infection [13].

Cell viability was assessed using the fluorescence intensity of the PI stain, as it serves as a reliable indicator of membrane damage and loss of integrity. To evaluate membrane permeability and leakage, the cells were stained with the membrane-impermeable PI dye. The relative viability of the cell suspensions was analyzed using a fluorescence microplate reader, and the mean fluorescence intensity was determined through spectrofluorometric evaluation, with each data point representing the average of ten measurements. The samples were excited at $480 \pm 20$ nm, and the integrated intensities of the emission from the suspensions were recorded in the green range ($530 \pm 12.5$ nm) and red range ($620 \pm 20$ nm). Treatment with ofloxacin exhibited the highest antibacterial activity per the increase in PI fluorescence, suggesting greater membrane damage (Figure 2).

### 3.3. Changes in Expression of Efflux Transporter Genes in P. aeruginosa during Biofilm Formation

To investigate the impact of biofilm formation triggers on the expression levels of the efflux transporter system MexAB-OprM, a comparison was made between their expressions in PAO1 planktonic cells and biofilms. Biofilms are well-known for their higher resistances to antibiotic treatments compared to planktonic cells [18]. The objective was to understand any alterations in the expression of the efflux transporter gene MexB in response to factors that induce biofilm formation, conferring increased resistance to both antibiotic inhibition and eradication.

The fold change values (mean of triplicate samples) were determined by comparing the transcription level of the MexB efflux transporter gene with that of the internal control RPSL. The expression of the MexB gene was assessed using qRTPCR to investigate its differential expression in planktonic and biofilm states. The results showed upregulation of MexB gene expression in the biofilm stage compared to the planktonic stage (Figure 3). This finding aligned with recent publications, indicating that a significant proportion of antibiotic-resistant clinical isolates of *P. aeruginosa* harbored the MexA and MexB genes, suggesting the crucial role of these active efflux pump systems in multi-drug resistances [17,19]. These studies also highlighted the contribution of the MexA and MexB genes to the elevated antibiotic resistance observed in biofilms formed by clinical isolates of *P. aeruginosa* [11]. In

contrast, the expression levels of the RPSL control gene remained consistent between the planktonic and biofilm stages.

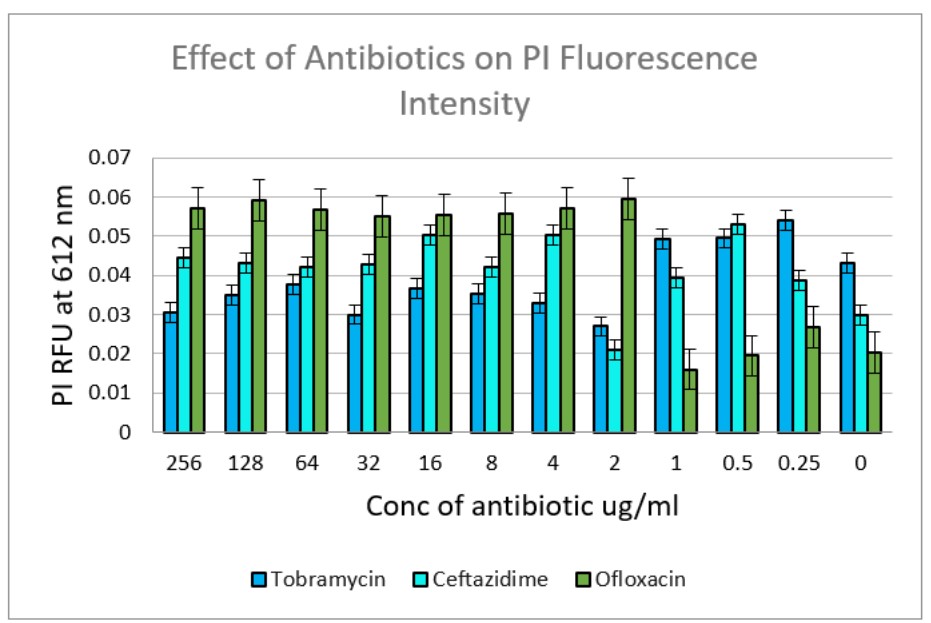

**Figure 2.** Effect of various antibiotics on the viability of PAO1 cells within a biofilm.

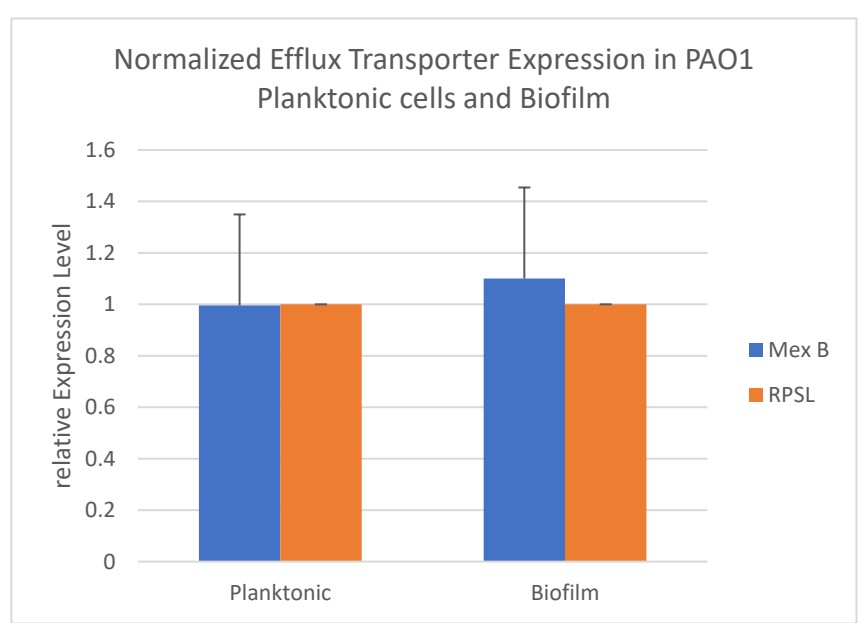

**Figure 3.** Normalized Expression of Efflux Transporter Mex B in PAO1 Planktonic Cells and Biofilm. Statistically highly significant as $p < 0.001$.

### 3.4. Effect of Various Antibiotics on Expression of Efflux Transporter Genes in Inhibition and Eradication Phases

The study aimed to investigate the impact of ceftazidime, ofloxacin, and tobramycin on the expression levels of the MexAB-OprM transporter systems in newly formed and mature PAO1 biofilms. The relative expression levels of MexB were compared after exposure to the antibiotics at their corresponding minimum inhibitory concentration (MIC) values for newly formed biofilms and minimum bactericidal concentration (MBC) values for mature biofilms. The objective was to assess whether these efflux pumps contributed to the reduced sensitivity to the tested antibiotics observed during and after biofilm formation.

Figure 4 represents the relative gene expression levels of MexB in PAO1 planktonic cells versus biofilms after treatment with the antibiotics at their determined MIC. Values represent fold changes (mean of triplicate samples) of the efflux transporter gene MexB in comparison with the transcription levels of the internal control RPSL. Statistical analysis using a paired *t*-test was performed to test if the mean MexB gene expression difference between the planktonic and biofilm cultures in the presence of various antibiotics was significant.

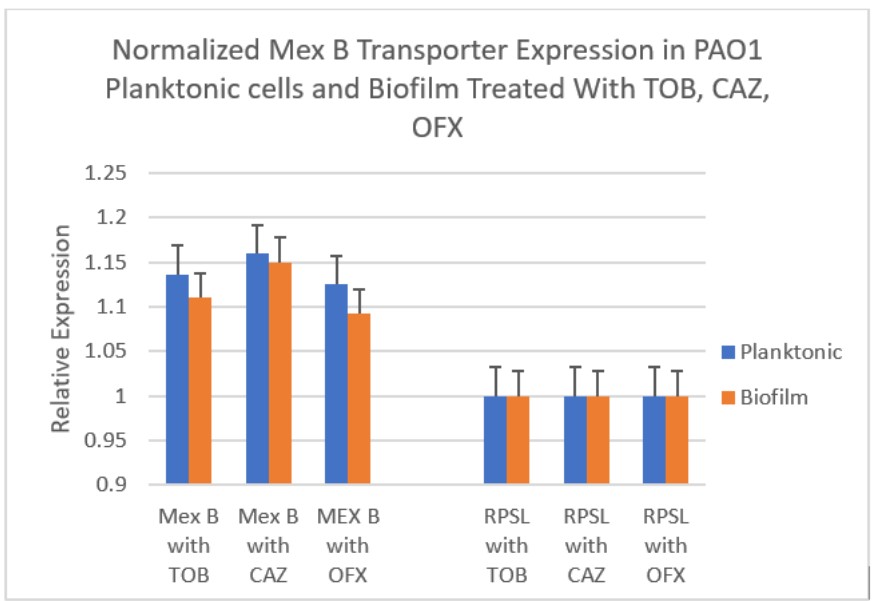

**Figure 4.** Relative Expression of Efflux Pump Gene MexB in PAO1 during (Planktonic) Inhibition of new biofilm formation and during (Biofilm) Eradication of mature biofilms after Treatments with three antibiotics, Tobramycin (TOB), Ceftazidime (CAZ), and Ofloxacin (OFX), at their Minimum Inhibitory Concentrations (MIC) for planktonic cultures and Minimum Bactericidal Concentrations (MBC) for biofilm cultures, respectively, as determined previously. Statistically highly significant as $p < 0.001$.

When comparing the impact of the tested antibiotics on the expression of the MexB gene in newly forming and mature PAO1 biofilms, it was observed that ceftazidime (CAZ) treatment led to a significant increase in the expression of the efflux transporter MexB gene relative to the internal control during both the inhibition (planktonic) and eradication (biofilm) phases. Previous studies have also reported the hyperexpression of MexAB-OprM genes, which are associated with increased resistance to cephalosporins [19,21], in *P. aeruginosa* biofilms, as evident from Figure 4. In contrast, treatment with ofloxacin and tobramycin resulted in much lesser increases in the expression of MexB compared to the expression of RPSL during both the inhibition and eradication phases (Figure 4). A paired *t*-test to determine the level of significance of the difference in MexB expression between planktonic and biofilm cultures showed a highly significant reduction in MexB expression in the PAO1 biofilm compared to the planktonic cultures in the presence of Ofloxacin and Tobramycin ($p < 0.001$), demonstrating their effectiveness in eradicating biofilm growth, whereas ceftazidime did not seem to produce a significant difference in the MexB expression level, showing that CAZ was not as effective as TOB and OFX. These findings supported the results obtained from studying the effectiveness of these antibiotics in inhibiting and eradicating *P. aeruginosa* biofilms, where ceftazidime exhibited the least efficacy with the highest MIC and MBC values (Figure 1A,B). This could have been attributed to the higher expression of the MexAB-OprM efflux pump system, which leads to increased extrusion of ceftazidime, thereby reducing its effectiveness. On the other hand, both ofloxacin and tobramycin demonstrated greater effectiveness compared to ceftazidime, with significantly

lower MexB expression levels. The MIC and MBC values also supported ofloxacin as the superior treatment option (Figure 1A,B). These findings were further supported by the lower expression levels of the efflux pump MexB gene observed after treating the PAO1 biofilms with ofloxacin under both inhibition and eradication conditions (Figure 4).

In cases of suspected *P. aeruginosa* infection, empirical antibiotic therapy typically involves the use of monotherapy or combination therapy [20,23]. Monotherapy options include β-lactam antibiotics or aminoglycosides. Combination therapy may involve the use of a β-lactam antibiotic (penicillin or cephalosporin) in combination with an aminoglycoside or the use of a carbapenem (imipenem or meropenem) in combination with antipseudomonal quinolones and an aminoglycoside [18,20,23].

A statistical comparison was conducted to determine whether data obtained for Mex B expression in Figures 3 and 4 were significant. A Student's *t*-test was completed to investigate the relationship between the alteration in MexB expression levels between planktonic cells and biofilms of PAO1 and the influence of antibiotics on the inhibition of the efflux pump. The statistical analysis revealed that there was a significant difference in MexB expression levels in all these cases. PAO1 biofilms expressed higher levels of MexB compared to planktonic cells (Figure 3). OFX and TOB treatment of PAO1 biofilms exhibited a significant decrease in MexB expression levels compared to planktonic cells with *p*-values of 0.02 and 0.008, respectively. Antithetically, CAZ demonstrated no substantial decrease in expression levels, as indicated by a *p*-value of 0.112.

The findings presented in this study also demonstrate a time-dependent efficacy of antibiotics against biofilm formation. The results, as depicted in Figure 1, indicate that a lower concentration of antibiotics is required to treat an early *P. aeruginosa* infection compared to a mature biofilm that has formed after 24 h. The study demonstrates that PAO1 biofilms are more vulnerable to antibiotics during the inhibition phase. This is supported by their heightened sensitivity to lower antibiotic concentrations in the early stages of formation. In contrast, during the eradication phase when biofilms have matured, they only exhibit sensitivity to much higher antibiotic concentrations [23]. These findings provide support for the use of monotherapy in empirical antibiotic therapy rather than combination treatments once the infection is confirmed. Preserving the effectiveness of antibiotics is a compelling reason to support monotherapy in the treatment of Pseudomonas aeruginosa infections. In addition to preventing drug interactions, minimizing side effects, and achieving cost-effectiveness, maintaining the efficacy of antibiotics is a crucial consideration. By utilizing a single antibiotic, the risk of promoting multidrug resistance and the emergence of resistant strains is reduced, ultimately preserving the effectiveness of individual antimicrobial agents in combating *P. aeruginosa* infections [24–26]. However, considering a transition to a more contemporary combination antibiotic therapy to address resistant strains of *P. aeruginosa* has the potential to enhance treatment outcomes [19,24,27]. The selection of therapy should be made on a case-by-case basis, considering various factors and individual patient characteristics [23,24]. Optimal decision-making in treatment selection necessitates a comprehensive understanding of all possible therapeutic options. Given the increasing prominence of *P. aeruginosa* emerging as a multi-drug resistant superbug and its prevalence, the identification of alternative antibiotics is vital.

To gain insights into the resistance mechanisms of *P. aeruginosa* PAO1 biofilms against the selected antibiotics, additional genomic analysis was performed to compare the expression levels of the genes responsible for the efflux pump proteins. This analysis aimed to identify any overexpressed or suppressed genes during antibiotic treatment. The results revealed that the MexB genes exhibited higher expression levels during the biofilm phase compared to the planktonic phase (Figure 3). Notably, when PAO1 biofilms were treated with ceftazidime, a significant upregulation of the MexB efflux gene was observed, in contrast to the treatment with ofloxacin and tobramycin, as shown in Figure 4. The observed upregulation of MexB may contribute to reduced antibiotic sensitivity by actively pumping the antibiotics out of the bacterial cells within the biofilm phase. These findings suggest

that MexB efflux pump could serve as a potential target for combating antibiotic resistance in the future.

## 4. Conclusions

Ofloxacin (OFX), Tobramycin (TOB) [21], and Ceftazidime (CAZ) have been widely used antibiotics that demonstrate effectiveness in treating *P. aeruginosa* infections. Among the three antibiotics tested during both the inhibition and eradication phases, OFX exhibited the highest efficacy in inhibiting the growth of *P. aeruginosa* strain PAO1. Conversely, CAZ displayed comparatively lower effectiveness in inhibiting the growth of *P. aeruginosa*. Although TOB exhibited slightly lower effectiveness compared to OFX, it still demonstrated greater efficacy than CAZ. Additionally, both tobramycin and ofloxacin were effective in eradicating biofilm growth. The significant reduction in MexB expression observed during biofilm treatment with OFX (Ofloxacin) and TOB (Tobramycin) compared to planktonic cells suggests a potential decrease in MexB-mediated efflux activity. This reduction implies that the biofilm cells may exhibit increased susceptibility to the OFX and TOB antibiotics, as efflux-mediated resistance mechanisms are attenuated. Consequently, the biofilm treatment with OFX and TOB antibiotics may be more effective in combating *P. aeruginosa* biofilm infections. In terms of inhibiting and eradicating *P. aeruginosa* biofilms, OFX (Ofloxacin) emerged as the most effective antibiotic overall, even at lower concentrations. On the other hand, CAZ demonstrated less effectiveness due to the observed resistance of the PAO1 strain in both the MIC and MBC phases. The lower effectiveness of CAZ in inhibiting biofilm growth may be related to the observed resistance of the PAO1 strain and may involve mechanisms other than MexB-mediated efflux. Additionally, TOB exhibited slightly lower efficacy compared to OFX but still demonstrated some effectiveness at relatively lower concentrations.

Furthermore, gene expression analysis was conducted to explore the potential role of the MexB efflux transporter in the observed antibiotic resistance during biofilm formation. The analysis revealed a higher expression of this gene in the PAO1 biofilms compared to planktonic cells, prior to antibiotic administration, and it was statistically significant ($p < 0.001$). This suggests that MexB efflux transporter may play a role in the biofilm-associated antibiotic resistance, as the upregulation of MexB expression in the biofilm stage indicates that the efflux pump is being more actively produced and potentially involved in the extrusion of antibiotics from the bacterial cells within the biofilm. This increased expression suggests that MexB may contribute to the reduced susceptibility of biofilm cells to antibiotics by facilitating the efflux of these antimicrobial agents. Therefore, MexB expression upregulation in biofilms implies a potential mechanism by which PAO1 biofilm cells can exhibit higher levels of antibiotic resistance compared to their planktonic counterparts [21]. The reduced expression of MexB in response to OFX and TOB treatment may contribute to the increased susceptibility of biofilm cells to these antibiotics. MexB expression levels can impact the effectiveness of antibiotics in inhibiting and eradicating *P. aeruginosa* biofilms, and the observed reduction in MexB expression supports the notion of MexB playing a role in efflux-mediated resistance. To evaluate the impact of MexB efflux pump on antibiotic susceptibility in PAO1 biofilms, genetic manipulation techniques like gene knockout or knockdown can be used to observe changes in susceptibility. These approaches provide insights into MexB's contribution to reducing antibiotic susceptibility in PAO1 biofilms. By neutralizing MexB activity via efflux pump inhibitors, intrinsic resistance could be reduced, rendering *P. aeruginosa* biofilms as more susceptible to antibiotics. While the MexB efflux pump may be an important factor in enhancing diverse antibiotic resistance, it is crucial to acknowledge that the development of resistance is a multifaceted process involving additional agents and mechanisms requiring study [28]. For example, quorum sensing utilized by *P. aeruginosa* to coordinate the expression of virulence factors and biofilm formation also holds the potential for further evaluation and manipulation in relation to efflux transporter gene expression [28–30].

Efflux pumps are a specialized tool of antibiotic resistance used by Pseudomonas aeruginosa to expel antibiotics. Previous studies have reported a 53.3% increase in the expression level of the *mexB* gene [30] and in at least one of the studied three genes of the MexAB-OprM efflux pumps in antibiotic-resistant *P. aeruginosa* isolates to show a significant correlation between the overexpression of these pumps and the frequency of multidrug resistance [30–33]. Our data has shown that MexB seems to contribute the most within the MexAB-OprM efflux pump system towards the antibiotic-resistance phenotype. Also, studies on multidrug resistant *P. aeruginosa* strains isolated from hospitals by Bialvaei et al. reported that the overexpression of *mexB* and *mexY* was significantly more prevalent in the studied strains, with a significant correlation with antibiotic resistance [7,33]. Altogether, the efflux pumps play important roles in increasing the resistance towards different antibiotics, but the role of other agents and mechanisms in the evolution of resistance should not be ignored. Limitations of this work include the lack of imaging studies to investigate the morphological changes brought about by antibiotics on the cells in the planktonic state and those embedded in the biofilms since fluorescence and confocal microscopy could not be performed due to unavailability of resources and equipment. Our earlier work showed no significant change in the appearances of the biofilms grown on microtiter plates and those of the PAO1 colonies grown on solid cultures [26]. The development of new antibiotics that can overcome the impact of efflux pumps remains a formidable challenge, one of many regarding *P. aeruginosa* MDR. Therefore, it is crucial to conduct additional studies to unravel the underlying mechanisms and establish the structure–function relationship of bacterial efflux systems [30,31]. Additionally, investigating the interactions between these pumps and other resistance mechanisms is highly recommended [31,32]. Such studies will provide valuable insights for the development of novel strategies to tackle antibiotic resistance effectively.

**Author Contributions:** Conceptualization, P.B.; methodology, E.K., R.G., W.L., S.P., R.M. and Y.P.; software, Y.P.; validation, R.G., E.K. and W.L.; formal analysis, E.K. and Y.P; investigation, E.K., R.G., W.L., S.P. and R.M.; resources, Y.P.; data curation, E.K. and P.B; writing—E.V and R.G.; writing—review and editing, P.B.; visualization, P.B.; supervision, P.B.; project administration, P.B.; funding acquisition, Y.P. All authors have read and agreed to the published version of the manuscript.

**Funding:** This research received no external funding.

**Institutional Review Board Statement:** Not applicable.

**Informed Consent Statement:** Not applicable.

**Data Availability Statement:** Data is contained within the article.

**Acknowledgments:** The authors are thankful to the Landers College of Men, Touro University, New York, USA, for offering laboratory facilities to carry out this study.

**Conflicts of Interest:** The authors declare no conflict of interest.

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
