# Peer review of "The Effect of Antibiotic Treatment and Gene Expression of Mex B Efflux Transporters on the Resistance in Pseudomonas Aeruginosa Biofilms"

_2673-8007, doi:10.3390/applmicrobiol3030049_

Round 1
Reviewer 1 Report
The study by Kello et al. investigated the resistance of Pseudomonas aeruginosa post treatments of several antibiotics. The authors also examined the expressions of a series of genes from Mex B efflux transporters. The authors claimed that Ofloxacin (along with Tobramycin) is the best in inhibiting the growth and biofilm formation of PAO1 strain in vitro. Importantly, the authors also proved that the higher expressions of examined gene clusters in biofilm may corelated to the antibiotic resistance in P. aeruginosa. Overall, the study is potentially interesting and shoud be acceptable after careful revisions and improvements in following sections.
1) This reviewer may miss the main logic of this study after reading the manuscript. What is the aim and why the authors only examined the gene family in this study. There has been already some study investigating the antibiotic resistance of this bacteria species and other related ones (especially transcriptome and functional genomics). Some more detailed background reviewing briefly should be better presented in introduction.
2) It seems this manuscript was prepared in a hash way. Some errors and typos are showing. A carefuly proofreading must be done before resubmission. e.g. the strain name (italic), not PA01, should be PAO1 if the authors used the correct one.
3) How about the morphologic changes post antibiotic treatments? Sometimes OD may not present the real growth of bacteria where the membrane fraction may affect the read under OD600.
4) In figure 2B (?, lower panel, labelling seems odd), the design of concentration test for CAZ and TOB are not very good. Between 2-4 ug/ml for TOB and 128-256 ug/ml for CAZ, more detailed analysis should be done.
5) The microscopic results should be also provided for Fig. 3 as main or supplemental figures.
6) The comparative results in figures 4 and 5 are not solid without statistical analysis.
7) How about of the significance of the results from this paper based on the findings from this study? The current ones seem a little overstated.
Some proofreading and polish should be performed.
Reviewer 2 Report
Minor revision
Please highlight the importance of the study.
Conclusions should be consice and brief.
Minor revisions
Reviewer 3 Report
The contents were worth to study and some useful conclusions were obtained. Reviewer recommends the publication of this paper in this journal. However, before acceptance, the following major concerns must be addressed:
1. Please reorganize the language of the Abstract, the first half is illogical and the data displayed in the abstract should be simplification.
2. The article’s language and grammar need to be optimized.
3. Authors showed biofilm formation from various experiments, however, cell morphology should be shown in detail from SEM and 2D CLSM for a comparison. Nowhere biofilms cell morphology is missing in this study
4. How do authors justify that PAO1 resistance in this study? There are a few works already published on PAO1 resistance and biofilm inhibition.
5. Authors missed many references to cite the PAO1 study, hence it has a lot of information on resistance mechanisms.
6. What is the synergistic role of these antibiotics in the resistance mechanism?
7. Matured biofilms must be taken into account after 48 h. 24h biofilm formation is not enough to produce big data.
8. Discuss more on the multidrug resistance mechanism for PAO1.
9. Conclusions are not satisfactory and too strong words. In particular, second part is too strong to conclude.
Missing articles should be noted
Round 2
Reviewer 1 Report
This review has no further comments on this revised manuscript.
Fine.